# Glutathione S-Transferases S1, Z1 and A1 Serve as Prognostic Factors in Glioblastoma and Promote Drug Resistance through Antioxidant Pathways

**DOI:** 10.3390/cells11203232

**Published:** 2022-10-14

**Authors:** Bo Cheng, Yu Wang, Abiola Abdulrahman Ayanlaja, Jing Zhu, Piniel Alphayo Kambey, Ziqiang Qiu, Caiyi Zhang, Wei Hu

**Affiliations:** 1Department of Psychiatry, The Affiliated Xuzhou Eastern Hospital of Xuzhou Medical University, Tongshan Road 379, Xuzhou 221000, China; 2The Key Lab of Psychiatry, Xuzhou Medical University, Tongshan Road 209, Xuzhou 221000, China; 3Department of Geriatric Psychiatry, The Affiliated Brain Hospital of Nanjing Medical University, Guangzhou Road 264, Nanjing 220029, China; 4Department of Neurology, Johns Hopkins University School of Medicine, 201 N Broadway, Baltimore, MD 21287, USA; 5Department of Neurobiology and Cell Biology, Xuzhou Medical University, Tongshan Road 209, Xuzhou 221000, China

**Keywords:** HPGDS, GSTZ1, GSTA1, glioblastoma, glutathione

## Abstract

The glutathione S-transferase (GST) family of detoxification enzymes can regulate the malignant progression and drug resistance of various tumors. Hematopoietic prostaglandin D synthase (HPGDS, also referred to as GSTS1), GSTZ1, and GSTA1 are abnormally expressed in multiple cancers, but their roles in tumorigenesis and development remain unclear. In this study, we used bioinformatics tools to analyze the connections of HPGDS, GSTZ1, and GSTA1 to a variety of tumors in genetic databases. Then, we performed biochemical assays in GBM cell lines to investigate the involvement of HPGDS in proliferation and drug resistance. We found that HPGDS, GSTZ1, and GSTA1 are abnormally expressed in a variety of tumors and are associated with prognoses. The expression level of HPGDS was significantly positively correlated with the grade of glioma, and high levels of HPGDS predicted a poor prognosis. Inhibiting HPGDS significantly downregulated GBM proliferation and reduced resistance to temozolomide by disrupting the cellular redox balance and inhibiting the activation of JNK signaling. In conclusion, this study suggested that HPGDS, GSTZ1, and GSTA1 are related to the progression of multiple tumors, and HPGDS is expected to be a prognostic factor in GBM.

## 1. Introduction

Glioblastoma (GBM) is one of the deadliest human cancers. It progresses rapidly and is highly resistant to chemotherapy agents [1,2,3,4,5,6,7,8]. Imbalances of the glutathione (GSH) redox system play key roles in tumorigenesis [9,10]. As a key enzyme that catalyzes the linking of GSH to cell-damaging factors such as chemotherapeutic drugs and peroxides, glutathione S-transferase (GST) has been found to affect the progression of a variety of cancers [11]. Prominent members of the GST family, such as GST pi 1 (GSTP1), tend to be highly expressed in various cancers. These enzymes promote tumor survival, proliferation, and drug resistance by inhibiting apoptotic pathways involving c-Jun N-terminal kinases (JNK), and autophagy pathways, such as those involving phosphoinositide 3-kinases [12,13,14,15,16,17]. 

Most existing studies of the roles of GSTs in cancer have tended to focus on GSTP1; therefore, the roles of other members of the GST family, such as hematopoietic prostaglandin D synthase (HPGDS/GSTS1), GST zeta 1 (GSTZ1), and GST alpha 1 (GSTA1), in the occurrence and development of GBM remain unclear. HPGDS is a member of the sigma subfamily of GSTs, and it can regulate the level of inflammation through its downstream product prostaglandin D2 (PGD2) [18]. PGD2 generated downstream of HPGDS has been shown to inhibit the occurrence of colon cancer by activating the prostaglandin D2 receptor [19,20]. In addition, deletion of the gene encoding HPGDS was found to induce pancreatic carcinogenesis and to increase the severity of lung adenocarcinoma in a mouse model [21,22]. 

GSTZ1 and GSTA1 have also been found to be associated with various forms of cancer. GSTZ1 decreases levels of reduced GSH in liver cancer tissues and thereby inhibits tumor progression [23,24]. It also promotes sorafenib-induced ferroptosis in liver cancer [25]. Conversely, Yogev found that high GSTZ1 expression is associated with poor prognosis in neuroblastoma [26]. In hepatocellular carcinoma, GSTA1 plays a tumor suppressor role via its impact on a signaling pathway involving AMP-activated protein kinase and mammalian target of rapamycin [27]. On the other hand, in lung cancer, inhibition of GSTA1 has been shown to induce apoptosis in chemotherapy-resistant cells [28]. 

To this point, research regarding HPGDS, GSTZ1, and GSTA1 has been limited mainly to their involvement in cancers of the digestive system. Considering the heterogeneity of tumors, it is important to further explore the role of GST-family enzymes in other systemic tumors, including GBM.

In this study, we hypothesized that HPGDS, GSTZ1, and GSTA1 are abnormally expressed in multiple cancers and may promote the malignant progression of GBM by regulating intracellular GSH levels, thus affecting the prognosis of patients.

## 2. Materials and Methods

### 2.1. Patient Data and Bioinformatics Analyses

This study used data from The Cancer Genome Atlas (TCGA) database (https://www.cancer.gov/tcga, last accessed date at 10 June 2022), the Gene Expression Omnibus (GEO) database (https://www.ncbi.nlm.nih.gov/geo/, last accessed date at 15 May 2022), and the Chinese Glioma Genome Atlas (CGGA) database [29] (http://www.cgga.org.cn/, last accessed date at 5 May 2022) to investigate clinical information related to HPGDS, GSTZ1, and GSTA1.

Gene expression analysis: This part was performed using TIMER2 (http://timer.cistrome.org/, last accessed date at 10 June 2022) and GEPIA2 (http://gepia2.cancer-pku.cn/#analysis, last accessed date at 20 July 2022). Protein levels of GST enzymes in tumor tissues were analyzed by UALCAN software based on the Clinical Proteomic Tumor Analysis Consortium dataset (http://ualcan.path.uab.edu/analysis-prot.html, last accessed date at 15 June 2022).

Survival analysis: Significance plots regarding the associations of overall survival (OS) and disease-free survival (DFS) with expression of GSTs in all TCGA tumors were obtained using the “Survival Plots” module in GEPIA2. Cutoff high (50%) and cutoff low (50%) values were used as expression thresholds to separate high and low expression cohorts. Hypothesis tests were performed using log-rank tests. 

SNP analysis: The frequencies of alterations, mutation types, and copy number alterations for each GST across all TCGA tumors were observed from cBioPortal (https://www.cbioportal.org/, last accessed date at 8 May 2022). The location of the most common mutation of each protein was displayed in a three-dimensional structural model by cBioPortal.

Cell immunity analysis: Analyses of the involvement of GST enzymes in tumor cell immunity were performed with the “Immune Genes” module of TIMER2. CD8^+^ T cells, CD4^+^ T cells, B cells, and regulatory T cells (Tregs) were selected for correlation analyses. Various algorithms, such as TIMER, CIBERSORT, CIBERSORT-ABS, QUANTISEQ, XCELL, MCPCOUNTER, and EPIC, were used for assessments of immune infiltration. The *p*-values and partial correlation values were obtained via the purity-adjusted Spearman’s rank correlation test. The data were visualized as a heatmap and a scatter plot.

Enrichment analysis: STRING was used to screen the top 50 interacting proteins associated with each GST protein, and GEPIA2 was used to obtain expression data from all TCGA tumors. The top 100 genes associated with each GST were identified based on expression. Pearson correlation analyses of each GST with their top 5 genes according to the expression were performed with the “Correlation Analysis” module of GEPIA2. Then, Kyoto Encyclopedia of Genes and Genomes (KEGG) and Gene Ontology (GO) analyses were performed with R (https://www.r-project.org/, last accessed date at 18 June 2022). The cellular functions of GSTs in GBM were also processed by Gene Set Enrichment Analysis (GSEA) using LinkedOmics (https://www.linkedomics.org/, last accessed date at 25 July 2022) [30].

The abbreviation and full name of cancers mentioned in this manuscript are shown in Appendix A. 

### 2.2. Cell Culture and Western Blotting

U251, U343, and U87 cell lines were purchased from the cell bank of the Chinese Academy of Sciences. The human astrocyte (HA) cell line was purchased from ScienCell Laboratory (Santiago, California, USA). All the GBM cells were cultured in high-glucose Dulbecco’s modified Eagle’s medium containing 10% fetal bovine serum and 1% penicillin and streptomycin (Beyotime, Shanghai, China). HA cells were cultured in Ham’s F-10 nutrient mixture medium containing 10% FBS and 1% penicillin and streptomycin. All the cell lines were cultured in incubators with 5% CO_2_ at 37 °C.

Following addition of HPGDS-inhibitor-1 (IN-1; 100 μM) (MCE, Shanghai, China), SP600125 (150 μM) (MCE, Shanghai, China), or Dimethyl sulfoxide (DMSO), cells were incubated in DMEM for 24 h. Cellular proteins were extracted and separated by gel electrophoresis and then transferred electrophoretically to a membrane. The membrane was blocked with a blocking reagent (Beyotime, Shanghai, China) for 15 min and washed 3 times with a washing buffer, and then incubated overnight with a primary antibody. The membrane was washed and then incubated with a fluorescently tagged secondary antibody for 2 h. Following a final wash, the membrane was scanned to determine the intensities of the bands. The intensities of the target protein bands were compared to that of the internal reference glyceraldehyde-3-phosphate dehydrogenase (GAPDH) in order to quantify the protein expression levels.

### 2.3. Cell Counting Kit 8 (CCK8) Assays

Cells were seeded (1 × 10^4^ cells/well) in 96-well plates and treated with inhibitors or drugs the next day. After being treated for 24 h, 20 μL of CCK8 reagent (Vicmed, Jiangsu, China) was added to each well, the plate was incubated for 30 min at dark, and the absorbance of cells at 450 nm were detected by a microplate reader.

### 2.4. Quantification of GSH and 4-Hydroxynonanol (4-HNE)

GSH was quantified with detection kit (Jiancheng, Nanjing, China) according to the manufacturer’s instructions. Briefly, 50 µL treated cells were collected and resuspend with 200 µL phosphate-buffered saline (PBS). Then, the cells were disrupted with an ultrasonic cell crusher at 3500 rpm for 10 min. The cell lysate and detection reagent 1 (100 µL), reagent 2 (10 µL), and reagent 3 (50 µL) were added to each well according to the manufacturer’s instructions. After mixing the contents, the absorbance at 405 nm was measured with a microplate reader.

The intracellular 4-HNE level was quantified using a detection kit (Jianglaibio, Shanghai, China). Briefly, 1 × 10^6^ cells were resuspended in 1 mL PBS, lysed with repeated freeze–thaw cycles, and centrifuged at 2000 rpm for 20 min. The enzyme-coated test plate was prepared with 40 μL diluent per well, and 10 μL of the clarified supernatant was added, followed by addition of 100 μL of the enzyme-labeled reagent. The plate was sealed with sealing film and incubated at 37 °C for 60 min. The sealing film was removed, the liquid was discarded, and each well was washed 5 times with a detergent solution. Color reagent A (50 μL) and B (50 μL) were added to each well. After gentle shaking and incubation at 37 °C for 15 min, the reaction was terminated with 50 μL of the stop solution. The absorbance of each well at 450 nm was then determined. 

### 2.5. Cell Proliferation Assay

Cell proliferation was detected by fluorescence staining with 5-ethynyl-2′-deoxyuridine (EdU). Briefly, cells were inoculated (1 × 10^4^ cells/well) in 96-well plates and cultured overnight. Then, the cells were treated with inhibitors on the next day. After treatment, the cells were incubated in a medium containing EdU (Ribo, Guangzhou, China) for an additional 2 h. After incubation, the cells were fixed for 30 min and stained with the EdU staining reagent Apollo (Ribo, Guangzhou, China), and 4′,6-diamidino-2-phenylindole (DAPI) (Sigma Aldrich, Shanghai, China) was used to counterstain the nuclei. The results were obtained with a fluorescence microscope. 

### 2.6. Apoptosis Assay

Apoptosis was measured via a TdT-mediated dUTP Nick-End Labeling (TUNEL) kit (KeyGen, Jiangsu, China). Briefly, the treated cells were fixed with 4% paraformaldehyde (KeyGen, Jiangsu, China) for 30 min and permeabilized for 5 min. TDT enzyme reaction solution (KeyGen, Jiangsu, China) was added to each well, and reactions were incubated in the dark at room temperature for 1 h. The cells were mixed with streptavidin labeling solution (KeyGen, Jiangsu, China) and incubated in the dark for 30 min. DAPI was added to counterstain the nuclei after washing the cells three times with PBS. After staining, the percentage of TUNEL-positive cells was counted. 

### 2.7. Data Analysis

Data were analyzed with SPSS 22.0 or GraphPad Prism 8 software. Data normality was tested using the Shapiro–Wilcoxon normality test, rejecting normality at *p* < 0.05. The significance of differences between groups were assessed by Student’s *t*-tests and one-way ANOVA tests. The survival distributions were estimated by Kaplan–Meier survival analysis, and the statistical significance between the stratified survival groups were assessed by the log-rank test. A two-sided *p* value < 0.05 was considered to indicate statistical significance. 

## 3. Results

### 3.1. HPGDS, GSTZ1, and GSTA1 Are Abnormally Expressed in Various Cancers

We first analyzed the expression levels of HPGDS, GSTZ1, and GSTA1 in human tumors in the TCGA database using TIMER2 and GEPIA2 software. As shown in Figure 1A,B, HPGDS exhibited a significantly decreased expression relative to the matched normal tissues in several cancer types, including bladder carcinoma (BLCA), breast cancer (BRCA), colon adenocarcinoma (COAD), lung adenocarcinoma (LUAD), and lung squamous cell carcinoma (LUSC), but it was significantly overexpressed in several other cancer types, including GBM, low-grade glioma (LGG), cholangio carcinoma (CHOL), kidney renal clear cell carcinoma (KIRC), kidney renal papillary cell carcinoma (KIRP), and thyroid cancer (THCA). The expression of GSTZ1 was significantly decreased in several cancers, including CHOL, COAD, KIRC, liver hepatocellular carcinoma (LIHC), and rectum adenocarcinoma (READ), while it was significantly overexpressed in GBM as well as LGG, diffuse large B-cell carcinoma (DLBC), kidney chromophobe (KICH), LUAD, and LUSC (Appendix A). The expression level of GSTA1 was decreased in most tumors, with significant decreases in GBM, BRCA, CHOL, COAD, head-neck squamous cell carcinoma (HNSC), and several other cancer types (Appendix A).

Analyses of the protein levels showed that levels of HPGDS were significantly lower in 7 out of 10 common tumors: BRCA, colon cancer, ovarian cancer (OV), uterine corpus endometrial carcinoma (UCEC), lung cancer, head and neck cancer, and liver cancer. Levels of HPGDS protein were significantly elevated in GBM and pancreatic cancer, which are consistent with the aforementioned HPGDS mRNA levels (Figure 1C). 

The level of GSTZ1 protein was also found to be decreased in BRCA, colon cancer, KIRC, pancreatic cancer, head and neck cancer, and liver cancer. Only OV was found to exhibit an increased GSTZ1 protein level. Interestingly, while we found that GSTZ1 mRNA was significantly increased in GBM (Appendix A), we noticed that the protein level of GSTZ1 was significantly reduced in this cancer type, suggesting that there may be abnormalities in the translation, modification, or protein stability of GSTZ1 in GBM. 

Unfortunately, the GSTA1 protein was undetectable in GBM and normal brain samples according to UALCAN, but the GSTA1 protein levels were found to be decreased in several tumor types, with only OV showing increased GSTA1 levels (Appendix A). Taken together, these results suggest that the expression levels of the GST family members are perturbed in various types of cancers. Specifically, the high expression of HPGDS in GBM indicate that this enzyme may have a tumor-promoting effect.

Further analysis showed that the expression of GSTs correlated with tumor grade. For example, HPGDS was differentially expressed in different stages of nine tumors, including those arising from adenoid cystic carcinoma (ACC), BLCA, KICH, and KIRC (Figure 2A). GSTZ1 and GSTA1 were also significantly differentially expressed in different stages of 7 and 5 tumor types, respectively (Appendix A). The expression levels of these GSTs may provide information regarding the grading of tumors in clinical diagnosis and treatment. Considering the specificity of the glioma grade, we further explored the correlation between GST expression and glioma grade.

### 3.2. Abnormally Expressed GSTs Are Significantly Correlated with the Prognosis of Patients with Various Cancers

To investigate the potential effects of the expression of GSTs on tumor progression, we divided samples into high-level GST expression and low-level GST expression groups, to analyze the correlations with tumor prognosis. We found that high levels of HPGDS were significantly correlated with decreases in overall survival (OS) in BLCA, LIHC, and OV patients, but high levels of HPGDS expression correlated with prolonged OS in KIRC, LUAD, and UCEC patients (Figure 2B). Considering the downregulation of HPGDS expression at the mRNA and protein levels in UCEC, HPGDS may play a role in inhibiting tumor progression in this cancer type (Figure 1). On the other hand, a high level of HPGDS expression was significantly correlated with low disease-free survival (DFS) in stomach adenocarcinoma (STAD) patients (Figure 2C). Specifically regarding glioma, although the HPGDS levels were found to have positive correlations with OS and DFS in patients with LGG and GBM, the correlations did not reach statistical significance (all *p* > 0.05). 

We also examined correlations of GSTZ1 and GSTA1 expression with patient prognosis. In contrast to HPGDS, high levels of GSTZ1 predicted higher OS in KIRP, KIRC, and OV patients and higher DFS in KIRC, LGG, and STAD patients (Appendix A). Similar to GSTZ1, high levels of GSTA1 were positively associated with higher OS and DFS in patients with ACC and were potentially associated with prolonged DFS in patients with LUSC, but levels of GSTA1 were only negatively correlated with OS in patients with skin cutaneous melanoma (SKCM) (Appendix A). Combined with the analysis of the mRNA levels of GSTs, as shown in Appendix A, it can be stated that GSTZ1 and GSTA1 may generally play a tumor suppressor role, while the role of HPGDS is more complex.

### 3.3. Single Nucleotide Polymorphisms Associated with Cancer Affect the Structure and Function of GSTs

In addition to differences in expression levels, changes to the protein structure and function induced by the presence of single nucleotide polymorphisms (SNP) may also lead to pathological changes. Therefore, we next compared the SNP status of GSTs in different tumor types in the TCGA database. These SNPs were mainly divided into four kinds, namely, mutation, amplification, deep deletion, and multiple alterations. SKCM was found to have the highest level of HPGDS SNPs. In esophageal squamous cell carcinoma (ESCA), CHOL, and LGG, the main SNP type of HPGDS was mutation, while in pheochromocytoma and paraganglioma (PCPG), the SNP type was only amplification. In GBM, both mutation and amplification alterations were identified (Figure 3A). 

SKCM was found to have the highest level of HPGDS SNPs. In esophageal squamous cell carcinoma (ESCA), CHOL, and LGG, the main SNP type of HPGDS was mutation, while in pheochromocytoma and paraganglioma (PCPG), the SNP type was only amplification. In GBM, both mutation and amplification alterations were identified (Figure 3A). The most frequent SNP site of HPGDS is the R193Q/^+^ site, but the SNP of this site is predicted to have little effect on the structure or function of the protein. This site is identified in yellow in the three-dimensional structure of the protein (Figure 3B–D). 

UCEC has the highest SNP level of the GSTZ1 gene, and the SNPs in DLBC, OV, ACC, PCPG, and ESCA are all amplification mutations. The most commonly identified SNP was found at the R21 site. This SNP is predicted to cause the missense mutation, which will strongly affect the structure and function of the GSTZ1 protein (Appendix A). 

The main SNP type of GSTA1 found in most cancers were amplification, while in some other cancer type, such as UCEC and COAD, the main SNP type found was mutation. The highest frequency of SNPs was found in the site coding for R131, and these SNPs are predicted to result in missense mutations. These missense mutations will lead to significant abnormalities in protein function (Appendix A).

### 3.4. GSTs Are Related to Tumor Immunity and Participate in Multiple Biological Processes

The tumor immune microenvironment is an important factor mediating tumor initiation, survival, and progression [31]. The GSH redox system and GSTs themselves have been found to affect tumor immune infiltration [32,33,34]. Here, we analyzed the correlation of the expression of GSTs with the level of immune infiltration of four tumor-associated immune cells: CD8^+^ T cells, CD4^+^ T cells, B cells, and Tregs. As shown in Figure 4A, HPGDS expression was significantly positively correlated with the level of B cells in LUSC and STAD; the highest correlation score obtained based on analyses using multiple algorithms is shown in Figure 4B. A significant positive correlation between HPGDS expression and infiltration by Tregs was detected in HNSC and THCA. However, we did not detect any consistent results regarding correlations of CD8^+^ T cell and CD4^+^ T cell infiltration with HPGDS expression. 

Regarding GSTZ1, we detected a negative correlation between the levels of infiltration of Tregs and GSTZ1 expression in THCA patients, and we did not detect any consistent correlations between infiltration by the other three immune cells and GSTZ1 levels (Appendix A). Similarly, there was no significant correlation identified between the expression of GSTA1 and infiltration by CD8^+^ T cells, CD4^+^ T cells, immune B cells, and Tregs in multiple types of cancer tissue.

In order to further clarify the role of abnormally expressed GSTs in tumors, we performed a screen using STRING software and identified 50 proteins that directly bind to HPGDS. The interaction network is shown in Figure 5A. In addition, we identified the top 100 genes in the TCGA database with the highest coefficients of correlation with HPGDS with GEPIA2. Surprisingly, we did not identify any genes that were present in the two sets of results, which would have suggested interactions at both the protein and gene levels (Figure 5C). There were significant positive correlations between HPGDS expression and the expression of the five most highly related genes (Figure 5B).

The KEGG pathway analysis of 150 genes from the two sets of results showed that HPGDS is mainly involved in a variety of energy metabolism pathways, including glucose metabolism and amino acid metabolism. It is also related to the hypoxia-inducible factor 1 pathway, which is an important pathway in the promotion of tumor progression [35]. This association may provide some explanation for the association between increased expression of HPGDS and worse patient prognoses in some cancers. 

The results of a GO enrichment analysis showed that the functions of HPGDS in biological processes were concentrated in glucose metabolism, especially glycolysis, suggesting that it may be related to the Warburg effect, which is a well-known alteration to metabolism in tumors. In terms of cell composition results of the GO analysis, HPGDS was found to be mainly involved in the movement of myofilaments and the cytoskeleton, which may suggest the role of HPGDS in migration and invasion of tumor cells. The main molecular function was found to be enzyme activity, especially its transferase activity. The catalysis of the addition of GSH moiety to other molecules may be the key cellular function of HPGDS in mediating tumorigenesis and progression (Figure 5D,E). 

The cellular functions of GSTZ1 tend to involve a variety of processes, such as energy metabolism and nucleic acid synthesis, while GSTA1 was found to be more closely involved with the metabolism of lipids. The results of the KEGG and GO analyses of GSTZ1 and GSTA1 are shown in Appendix A.

### 3.5. High HPGDS Expression Predicts Poor Prognosis in Glioma Patients

The aforementioned pan-cancer analyses demonstrated that HPGDS has a higher prognostic value in glioma than do GSTZ1 and GSTA1. Analysis of the CGGA database showed that the expression level of HPGDS in high-grade gliomas was significantly higher than that in low-grade gliomas, suggesting that the HPGDS expression levels were positively correlated with tumor grade (Figure 6A). Further analysis of patient prognosis showed that patients with higher HPGDS mRNA levels had significantly worse prognoses (Figure 6B). This trend was consistent with trends identified using the TCGA database, as analyzed in Figure 2. However, the difference obtained using the China-specific database was more significant than that for the global database, suggesting that regional and ethnic differences may influence the correlation between HPGDS and glioma prognosis. 

The patient samples were further divided into grade 2 gliomas, grade 3 gliomas, and grade 4 gliomas for comparative analysis. It was found that there was a significant negative correlation between HPGDS and prognosis in patients with grade 2 and grade 3 gliomas. Although the survival time of patients with high HPGDS expression was also shorter in grade 4 glioma, the difference was not statistically significant, which may be related to the poor overall prognosis of grade 4 glioma patients (Figure 6B).

Gliomas with mutations in the isocitrate dehydrogenase 1 (IDH1) gene and the 1p/19q co-deletion have a better prognosis [36]. A comparison of the expression level of HPGDS with the mutation status of IDH1 showed that HPGDS was more highly expressed in IDH1 wild-type gliomas, especially grade 2 and 4 gliomas. Similarly, HPGDS was also expressed highly in grades 2, 3, and 4 gliomas lacking the 1p/19q co-deletion, further suggesting that high levels of HPGDS predict poorer patient outcomes (Figure 6C,D). 

When analyzing similar data regarding GSTZ1 in the CGGA database, we did not obtain the results that suggested any significant correlations between GSTZ1 expression and glioma status (Appendix A). In addition, there are not sufficient GSTA1-related data in the CGGA database to allow robust analyses.

To clarify the impact of HPGDS on GBM progression, we further performed functional analyses based on GBM sample data in the TCGA database. The genes highly related to HPGDS in GBM were screened with LinkedOmics software, and the KEGG pathway analysis data based on GSEA showed that HPGDS was related to cellular functions involving phagosomes and lysosomes in GBM. The GO analysis results based on GSEA showed that HPGDS is related to the neutrophil-mediated immunity of GBM, secreted granule membranes, cytokine receptor activity and cytokine binding, and other functions, and it can participate in the regulation of apoptosis (Appendix A).

### 3.6. Inhibition of HPGDS Lowers GSH Levels and Decreases Proliferation of GBM Cells 

To further characterize the expression of HPGDS in GBM, we first detected the protein level of HPGDS in a human astrocyte cell line (HA) and three GBM cell lines (U251, U343, and U87) (Figure 7A). Consistent with the results of the bioinformatics analyses, the protein levels of HPGDS were significantly elevated in the GBM cell lines. We further examined the effect of HPGDS on the GSH levels in GBM by treating U251 and U87 cells with a potent inhibitor of the enzyme activity of HPGDS, IN-1. The results showed that inhibition of HPGDS led to lower GSH levels in both cell types and resulted in the accumulation of intracellular 4-HNE, a byproduct of lipid peroxidation (Figure 7B,C). Using CCK8 and EdU assays to quantify proliferation upon treatment with IN-1 or the vehicle control demonstrated that inhibition of HPGDS can also significantly reduce the proliferation of GBM cells (Figure 7D–G).

### 3.7. Inhibition of HPGDS Reduces Drug Resistance in GBM

GBM is highly resistant to chemotherapeutic agents. Here, we examined the effect of HPGDS on the level of drug resistance in GBM. CCK8 analyses were performed on cells pre-treated with IN-1 or the vehicle control prior to treatment with the chemotherapeutic agent temozolomide (TMZ) (200 μM) for 24 h. In cells treated with the inhibitor, the cytotoxicity of TMZ gradually increased with time and was significantly higher than that of the control group. Similarly, the results of the TUNEL assays showed that compared with the control group, the rate of apoptosis of the GBM cells treated with TMZ was significantly increased after pretreatment with IN-1 (Figure 8).

### 3.8. HPGDS Mediates Drug Resistance of GBM via the JNK Pathway

The JNK pathway has been found to interact with multiple GST enzymes, including GSTP1 [37,38,39], and these interactions have been shown to promote apoptosis in glioma [40,41,42]. We thus hypothesized that the enhancing of drug resistance in GBM by HPGDS may be partially dependent on its inhibition of JNK pathway activation. 

Western blot results showed that inhibition of HPGDS with IN-1 resulted in a significant upregulation of JNK phosphorylation (Figure 9A,C). To determine whether HPGDS suppresses apoptosis by inhibiting the activation of the JNK pathway, we treated cells with TMZ after a combined treatment with IN-1 and the JNK inhibitor SP600125. The results of the CCK8 and TUNEL assays showed that the enhanced cytotoxicity and apoptosis of TMZ induced by IN-1 were reversed upon combined treatment with SP600125, indicating that HPGDS promotes drug resistance of GBM by inhibiting the activation of the JNK pathway (Figure 9B,D–F).

## 4. Discussion

In this study, in order to gain an in-depth understanding of the roles of HPGDS, GSTZ1, and GSTA1—the less-studied members of the GST family—in tumorigenesis and development, we first performed a pan-cancer bioinformatics analysis. The results showed that the expression levels of HPGDS were significantly increased or decreased in a variety of cancers. In contrast, GSTZ1 and GSTA1 generally showed low expression in tumor tissues, which suggests that these enzymes mainly exhibit tumor suppressive effects, and that the reduction in GSTZ1 and GSTA1 levels is associated with poor prognosis of KIRC, ACC, and other tumors [23,25,27].

We then selected HPGDS for in-depth research into its potential as a prognostic factor for GBM. Previous research on HPGDS has mainly focused on cancers of the digestive system, and it has been studied in large part for the role of HPGDS in the production of the anti-inflammatory factor PGD2. HPGDS expression in the gut is induced by reactive oxygen species, and through its product, PGD2, and metabolite, 15d-PGJ2, it antagonizes the action of the pro-inflammatory factor PGE2, downregulates interleukin-1β and macrophage inflammatory protein 2, and can induce inflammation [18,19,20,21]. It can also regulate the level of GSH in cells through PGD2-dependent activation of the Nrf2 pathway, thereby maintaining intestinal homeostasis and inhibiting the occurrence and progression of tumors [43,44,45].

Consistent with previous results, we detected low expression of HPGDS in colorectal cancers in this study. Other studies have also shown that HPGDS can affect the progression of OV, lung cancer, gastric cancer, and BRCA, among other cancer types [46,47,48,49]. In our study, a TCGA-based analysis also yielded consistent results in that HPGDS was found to exhibit significant reductions at the mRNA and protein levels in these cancer types. This consistency suggests that our bioinformatics approach was reliable.

It is worth noting that in a study on LGG, researchers found that HPGDS may act as a downstream target of miR-326 to promote the proliferation of LGG and regulate tumor apoptosis through the arachidonic acid metabolic pathway [50]. We therefore hypothesized that HPGDS may be involved in mediating the malignant progression of gliomas. The results from our database analyses support this hypothesis. There is an increase in the mRNA and protein levels of HPGDS in GBM, and a high expression of HPGDS was also detected in LGG. The results based on analyses of TCGA data further showed that patients with high expression of HPGDS have poor prognoses. Although the results from TCGA did not reach the level of statistical significance, when we analyzed the CGGA data, we detected a significant correlation between high HPGDS and the malignancy grade and prognosis of Chinese glioma patients. Factors such as ethnicity, environment, and dietary habits may have contributed to this disparate outcome. Further analysis found that patients with wild-type IDH1 and 1p/19q non-deletion had higher HPGDS levels, suggesting that elevated HPGDS levels predict a worse prognosis for patients [36]. 

GSEA analysis of HPGDS in GBM further revealed its diverse biological functions in GBM. Subsequent experiments in cultured cells showed that HPGDS plays a role in regulating cell proliferation in GBM similar to its role in LGG [50], and the use of an HPGDS-specific inhibitor significantly reduced the level of proliferation in two GBM cell lines. We speculate that this effect of inhibition may be related to ferroptosis caused by an imbalance in GSH levels, which has been shown to inhibit cell proliferation in GBM [51]. The effect of HPGDS inhibition on ferroptosis will be further explored in follow-up studies. 

The GSH redox system plays an important role in cell survival and apoptosis [52,53,54], and inhibition of HPGDS was found to result in a downregulation of the GSH levels and a reduction in cellular drug resistance. Considering that the JNK pathway plays an important role in promoting apoptosis in glioma, and a variety of GSTs can regulate the activation of the JNK signaling pathway [42,55,56], we speculate that the resistance-promoting function of HPGDS is partly attributable to the inhibition of apoptosis resulting from downregulated activation of the JNK pathway. In this study, inhibition of HPGDS led to significant activation of JNK phosphorylation, and interference with the activation of the JNK pathway reversed the reduction in cell drug resistance caused by inhibition of HPGDS, indicating that HPGDS functions similarly to other GSTs in regulating the activation level of apoptotic pathways.

## 5. Conclusions

We explored the biological roles of three currently understudied GSTs in tumors. GSTZ1 and GSTA1 are expressed at low levels in a variety of tumors and may play a role in tumor suppression. HPGDS plays a role in promoting tumor progression by regulating the level of cellular GSH and the activation of the JNK pathway in GMB. It may be used as a new factor to evaluate the prognosis of GBM patients.

## Figures and Tables

**Figure 1 cells-11-03232-f001:**
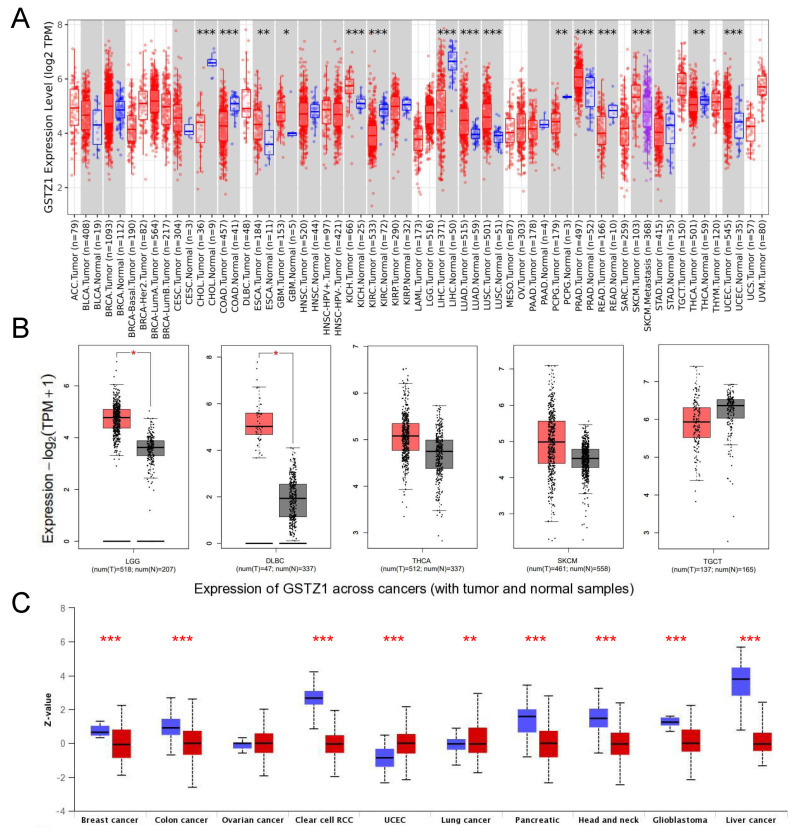
Bioinformatics analysis of the HPGDS mRNA and protein levels in different tumors. (**A**,**B**) The expression of HPGDS at the mRNA level in the noted cancer types was analyzed with TIMER2. (**C**) The expression of HPGDS at the protein level in the noted cancer types was analyzed with UALCAN, the red lines mean cancer group while the blue lines mean the normal group. * *p* < 0.05, ** *p* < 0.01, *** *p* < 0.001, ns: *p* > 0.05.

**Figure 2 cells-11-03232-f002:**
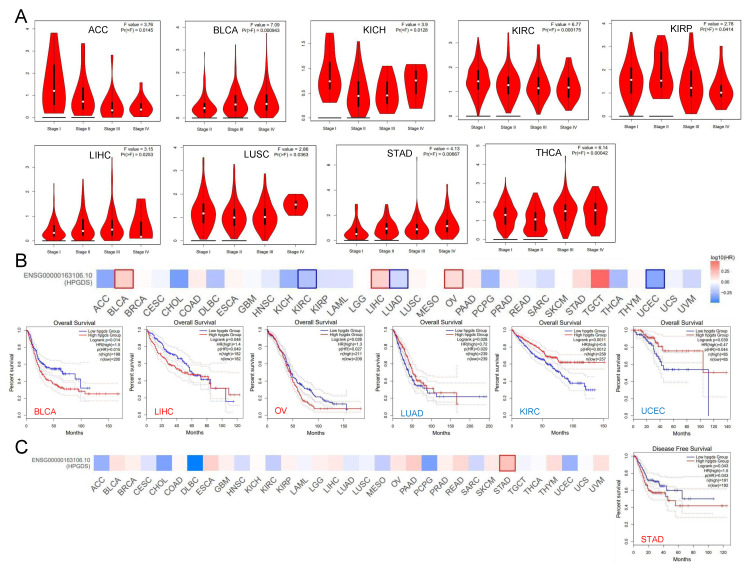
Analysis of the TCGA database to investigate correlations between expression of HPGDS and cancer prognosis. (**A**) The expression level of HPGDS was analyzed according to the main pathological stages (stage I, stage II, stage III, and stage IV) of different cancers. The equation Log2(TPM + 1) was applied to calculate the log scale. (**B**,**C**) The OS and DFS of different cancers according to the TCGA database were analyzed relative to expression of HPGDS.

**Figure 3 cells-11-03232-f003:**
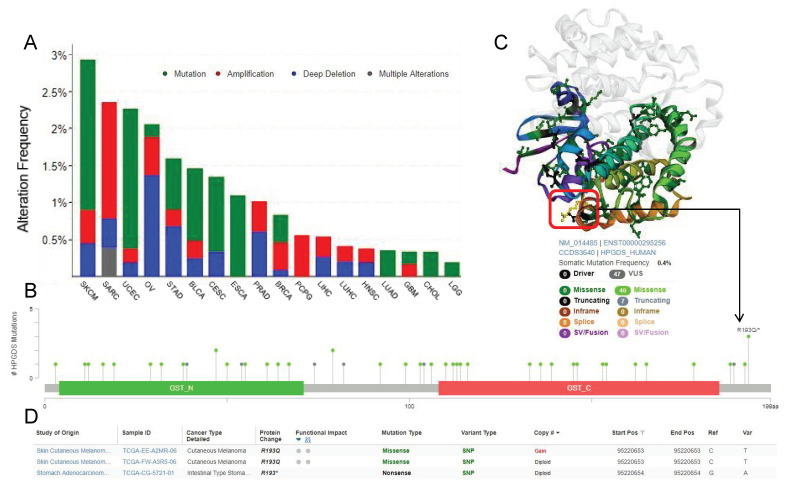
Analysis of the TCGA database for the mutation status of HPGDS in different cancers. (**A**) The SNP level of HPGDS in TCGA tumors. (**B**) The SNP level of each site of HPGDS in TCGA tumors. (**C**,**D**) Information regarding the mutation site with the highest SNP frequency (R193Q/^+^). The location of this site in the three-dimensional structure of HPGDS is shown in yellow.

**Figure 4 cells-11-03232-f004:**
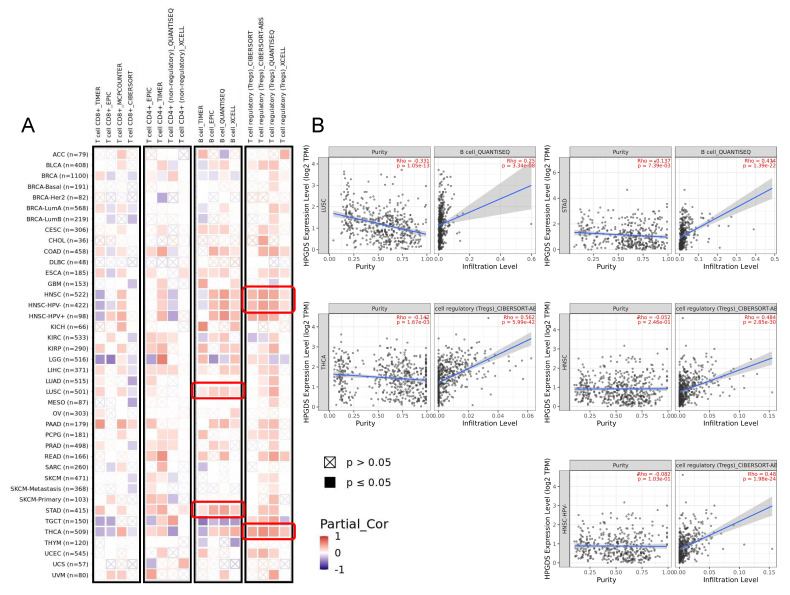
Analysis of the correlation between HPGDS expression and infiltration of cancer-associated immune cells. (**A**) Multiple algorithms were used to explore potential correlations between the expression of HPGDS and the infiltration of cancer−associated CD8^+^ T cells, CD4^+^ T cells, B cells, and Tregs across all types of cancer in TCGA. (**B**) The results with the highest correlation scores, as verified by multiple algorithms, are shown.

**Figure 5 cells-11-03232-f005:**
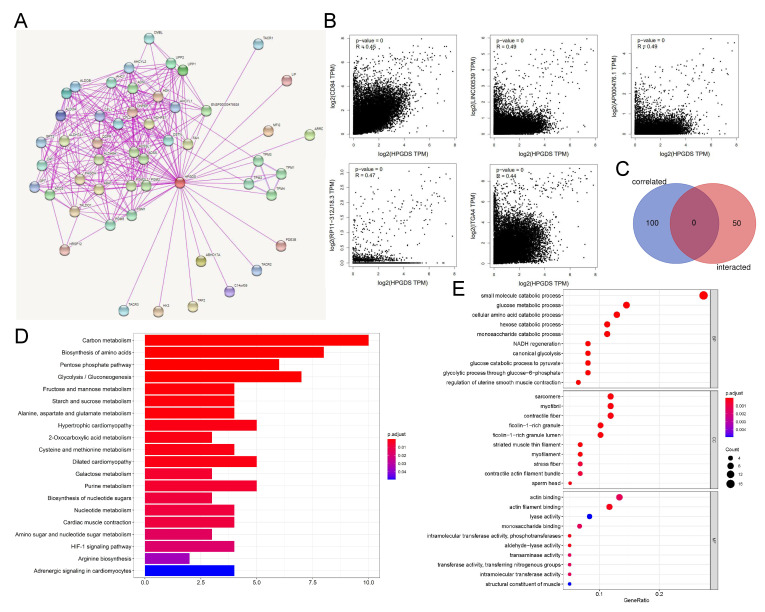
HPGDS-related gene enrichment analysis. (**A**) Experimentally determined HPGDS-binding proteins were identified using STRING. (**B**) The correlations of the expression of HPGDS and the five most related genes in cancers are shown. (**C**) An analysis of the intersections of genes encoding HPGDS-binding proteins and expression-correlated genes was conducted. (**D**) KEGG pathway analyses were performed on the genes encoding HPGDS-binding proteins and expression-correlated genes. (**E**) GO analyses were also performed.

**Figure 6 cells-11-03232-f006:**
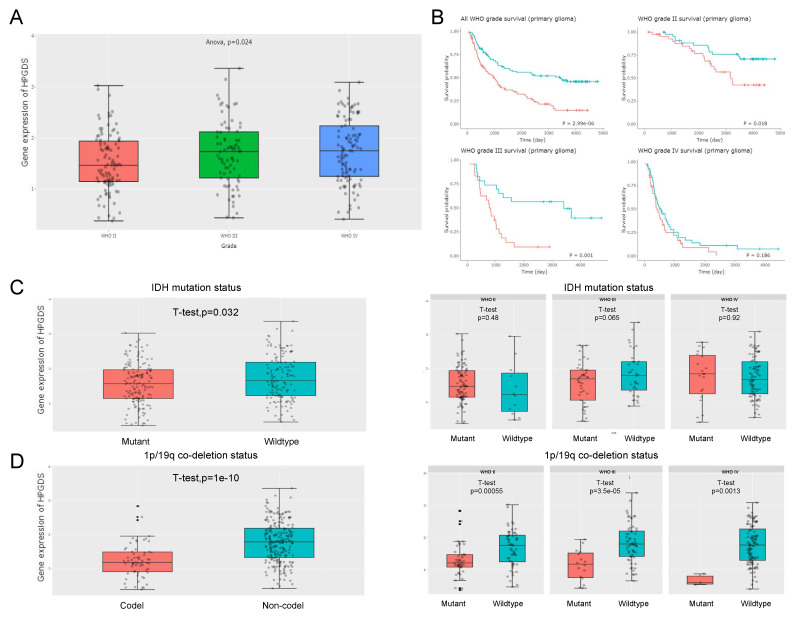
A correlation analysis of HPGDS expression and patient prognosis based on the CGGA database. (**A**) The results of CGGA database analyses showed that the level of HPGDS was induced more strongly in high-grade glioma (HGG) than in low-grade glioma (LGG). (**B**) The survival rates of patients with high HPGDS expression is significantly lower than patients with lower HPGDS expression, the red line mean high HPGDS expression group, the blue line mean low HPGDS expression group. This trend also existed in grade 2 and 3 gliomas, but there was no significant difference in grade 4 gliomas. (**C**) The results of the CGGA database analyses showed that HPGDS was more highly expressed in IDH1 wild-type glioma; the difference was significant in grade 3 glioma. (**D**) HPGDS was highly expressed in 1p/19q non-deleted glioma, and the difference was significant in grade 2, 3, and 4 gliomas.

**Figure 7 cells-11-03232-f007:**
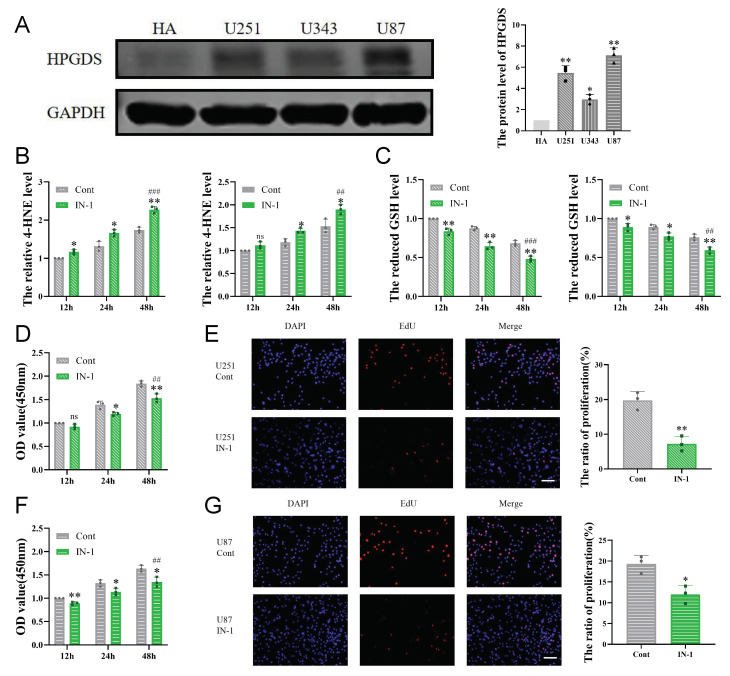
HPGDS is highly expressed in GBM and affects the cellular redox status and proliferation. (**A**) The levels of HPGDS protein in GBM cell lines are significantly higher than in normal glial cell lines. (**B**,**C**) Inhibition of HPGDS resulted in an accumulation of 4-HNE and a decrease of intracellular GSH in GBM cells. (**D**–**G**) The effects of inhibition of HPGDS on the proliferation of U251 and U87 cells were detected by CCK8 and EdU assays. * *p* < 0.05, ** *p* < 0.01; ## *p* < 0.01, ### *p* < 0.001, ns: *p* > 0.05.

**Figure 8 cells-11-03232-f008:**
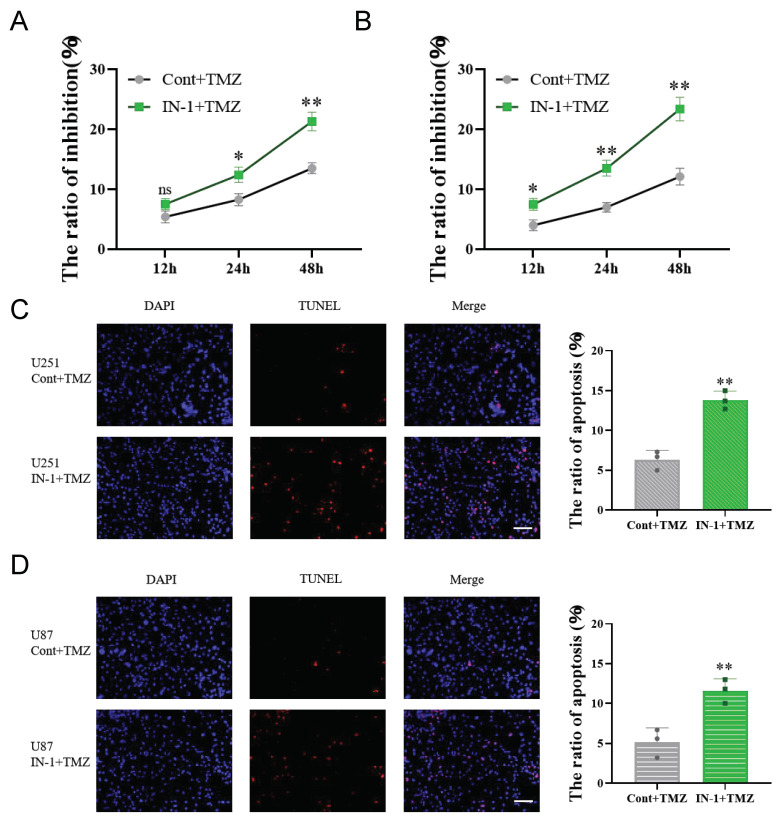
Inhibition of HPGDS reduces drug resistance of GBM cells. (**A**,**B**) Inhibition of HPGDS increases inhibition of U251 and U87 cells by TMZ. (**C**,**D**) Inhibiting HPGDS increases TMZ-induced apoptosis of U251 and U87 cells.* *p* < 0.05, ** *p* < 0.01, ns: *p* > 0.05.

**Figure 9 cells-11-03232-f009:**
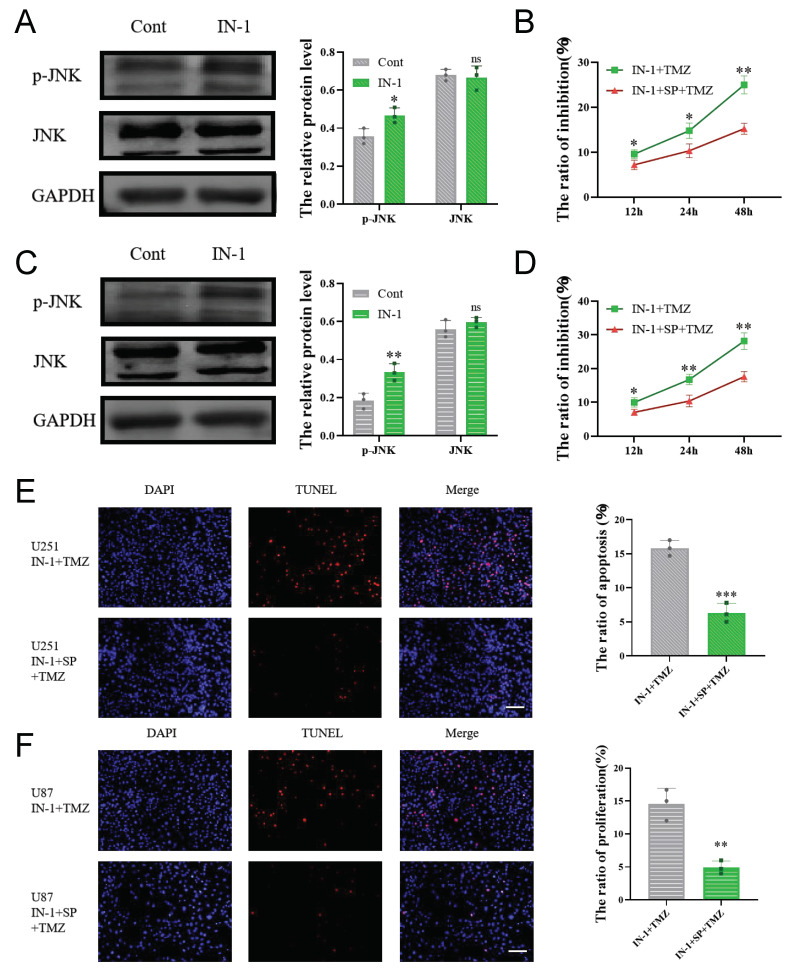
HPGDS promotes drug resistance in GBM by inhibiting JNK activation. (**A**,**C**) Inhibiting HPGDS significantly inhibits JNK phosphorylation. (**B**,**D**) CCK8 detection of the level of inhibition induced by SP600125 in combination with IN-1 and TMZ in U251 and U87 cells. (**E**,**F**) TUNEL results, showing that inhibiting JNK activation reduced the level of apoptosis induced by treatment with IN-1 and TMZ. * *p* < 0.05, ** *p* < 0.01, *** *p* < 0.001, ns: *p* > 0.05.

## Data Availability

The data is contained within the article or Appendix A.

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
