# Peer review of "Glutathione S-Transferases S1, Z1 and A1 Serve as Prognostic Factors in Glioblastoma and Promote Drug Resistance through Antioxidant Pathways"

_cells, 2022, doi:10.3390/cells11203232_

Round 1
Reviewer 1 Report
The present manuscript deals with a very interesting and topical issue and it is well written. A comprehensive evaluation of data from genetic databases will allow a comprehensive assessment of the available results. I have only few comments and suggestion for improving of this manuscript.
· Line 22 – founnd
· Line 67 – do not put results in the aims of your study
· Add hypothesis in your manuscript
· All abreviation have to be explained at first mention (e.g. line 118 – DMSO, line 125 – GAPDH)
· Determination of GSH (line 133-139) – describe the method of GSH analysis more in detail (e.g. pH of buffer, used volumes)
· Figures – Description of some figures is missing (median, mean, SD, quartiles, extreme values? etc)
· All abbreviations in Figures have to be explained
Author Response
Dear reviewer:
Thanks for review, here is the response to the comments.
- Line 22 – founnd:founnd to found
- Line 67 – do not put results in the aims of your study:The results have beed deleted.
- Add hypothesis in your manuscript:The hypothesis has been added at Line 67-69.
- All abreviation have to be explained at first mention (e.g. line 118 – DMSO, line 125 – GAPDH):All abbreviations have been explained (figure S13) .
- Determination of GSH (line 133-139) – describe the method of GSH analysis more in detail (e.g. pH of buffer, used volumes):The description has been modified.
- Figures – Description of some figures is missing (median, mean, SD, quartiles, extreme values? etc):Thanks for pointing this out. All authors agreed that more description of the figures would have contributed to a better presentation of the findings. However, given that each figure contains many results, the additional description of the figures may confuse readers. Therefore, we hope that no additional numerical descriptions will be added to this manuscript.
- All abbreviations in Figures have to be explained:All abbreviations have been explained (figure S13) .
Reviewer 2 Report
In their manuscript Cheng et al. explore the role of HPGDS, GSTZ1 and GSTA1 in tumorigenesis and progression of multiple cancers. Indeed, these are three members belonging to the GST family and their abnormal expression and/or involvement in tumor progression and drug resistance have been already studied, especially in tumors of digestive system. The authors, by means of bioinformatic tools, analyzed the enzymes expression and they found an important prognostic value in several cancers. In detail, expression level of GSTS1 positively correlated with the grade of glioma; so the authors, performing in vitro experiments, studied its involvement in malignant progression and in chemoresistance of GBM. Since GBM is an aggressive tumor, showing high heterogeneity, it’s of great clinical significance discovering new factors playing a key prognostic function. The study design is clear, the manuscript is precise and well written, with comprehensive literature data.
I only have a major point to mention: in figure 1 there is a misunderstanding. Both in the text and in the legend the authors discuss about HPGDS analysis; instead, figure 1 shows GSTZ1 analysis and, as a consequence, figure 1 corresponds to figure S3.
Author Response
Dear review:
Thanks for review ,here is the response to the comments:
- We have change the figure S3(Line 219-210) to figure 1 in the manuscript.